# A Comprehensive Analysis of FUT8 Overexpressing Prostate Cancer Cells Reveals the Role of EGFR in Castration Resistance

**DOI:** 10.3390/cancers12020468

**Published:** 2020-02-18

**Authors:** Naseruddin Höti, Tung-Shing Lih, Jianbo Pan, Yangying Zhou, Ganglong Yang, Ashely Deng, Lijun Chen, Mingmimg Dong, Ruey-Bing Yang, Cheng-Fen Tu, Michael C. Haffner, Qing Kay Li, Hui Zhang

**Affiliations:** 1Department of Pathology, Johns Hopkins School of Medicine, Baltimore, MD 21287, USA; tlih1@jhmi.edu (T.-S.L.); jpan16@jhmi.edu (J.P.); yzhou122@jhu.edu (Y.Z.); glyanglife@hotmail.com (G.Y.); ashleysdeng@gmail.com (A.D.); lchen105@jhmi.edu (L.C.); mdong7@jhmi.edu (M.D.); michael.c.haffner@gmail.com (M.C.H.); qli23@jhmi.edu (Q.K.L.); hzhang32@jhmi.edu (H.Z.); 2Institute of Biomedical Sciences, Academia Sinica, Taipei 11529, Taiwan; rbyang@ibms.sinica.edu.tw (R.-B.Y.); pattukimo@yahoo.com.tw (C.-F.T.); 3Sidney Kimmel Comprehensive Cancer Center, Johns Hopkins University School of Medicine, Baltimore, MD 21231, USA

**Keywords:** prostate cancer, alpha (1,6) fucosyltransferase, proteomics, glycoproteomics

## Abstract

The emergence of castration-resistance is one of the major challenges in the management of patients with advanced prostate cancer. Although the spectrum of systemic therapies that are available for use alongside androgen deprivation for treatment of castration-resistant prostate cancer (CRPC) is expanding, none of these regimens are curative. Therefore, it is imperative to apply systems approaches to identify and understand the mechanisms that contribute to the development of CRPC. Using comprehensive proteomic approaches, we show that a glycosylation-related enzyme, alpha (1,6) fucosyltransferase (FUT8), which is upregulated in CRPC, might be responsible for resistance to androgen deprivation. Mechanistically, we demonstrated that overexpression of FUT8 resulted in upregulation of the cell surface epidermal growth factor receptor (EGFR) and corresponding downstream signaling, leading to increased cell survival in androgen-depleted conditions. We studied the coregulatory mechanisms of EGFR and FUT8 expression in CRPC xenograft models and found that castration induced FUT8 overexpression associated with increased expression of EGFR. Taken together, our findings suggest a crucial role played by FUT8 as a mediator in switching prostate cancer cells from nuclear receptor signaling (androgen receptor) to the cell surface receptor (EGFR) mechanisms in escaping castration-induced cell death. These findings have clinical implication in understanding the role of FUT8 as a master regulator of cell surface receptors in cancer-resistant phenotypes.

## 1. Introduction

Prostate cancer remains one of the most common malignancies in aging men and is the second leading cause of cancer related deaths in the USA [1]. Autopsy studies have confirmed that around 60%–70% of men in their 7th and 8th decade of life have some histological evidence of prostate cancer [2,3]. While the majority of men with prostate cancer will develop microscopic disease, only a few of these patients will develop aggressive prostate cancer [4]. Clinically, organ-confined prostate cancer is managed surgically by radical prostatectomy or by radiation therapy [5], however, in some patients with disease recurrence or those diagnosed with advanced high-risk prostate cancer patients, androgen-deprivation therapy (ADT) serves as the primary treatment selection [6]. ADT has been shown to reduce tumor burden and improve clinical outcome; however, long-term ADT leads to resistance and progression to CRPC which is fatal [7]. Several mechanisms of resistance have been linked to the development of castration-resistant phenotypes including the gain of function mutations or amplification of the *AR* gene, the emergence of AR splice variants, overexpression of the AR cofactors, and ligand-independent activation of the AR by growth factors. In addition to these well-established mechanisms, alterations in post-translational modifications including glycosylation have been recently recognized in supporting cancer cells proliferation.

Glycosylation is recognized as one of the most common modifications on proteins and has been linked to play important roles in many cellular processes [8]. Aberrant fucosylation that results from the deficiency or overexpression of fucosyltransferases (FUTs) is associated with a variety of human diseases, including cancer [9,10]. Unlike other members of the fucosyaltrasferases that are functionally redundant, the α (1,6) fucosyltransferase (FUT8) is the only enzyme responsible for the α 1,6-linked (core) fucosylation of proteins, conjugating a fucose sugar to the inner most moiety of the N-linked glycans [7]. Many studies suggest the importance of core fucosylation in regulating protein trafficking and functions within and outside the cells [11,12]. Transgenic animal models have also been explored to evaluate the role of core fucosylation [13,14]. Ectopic expression of FUT8 in animal models have resulted in the steatosis-like phenotype in transgenic mice [15], on the other hand knocking out FUT8 in mice was reported to dramatically decrease the postnatal survival of the pups [14]. Similarly, core fucosylation is known to play important roles in the ligand-binding affinity of transforming growth factor (TGF)-β1 receptor, epidermal growth factor (EGF) receptor [16], and integrin α3β1 [17]. Loss of the core fucose on these receptors leads to a significant reduction in ligand-binding ability and downstream signaling activity. Furthermore, an increase in core fucosylation on E-cadherin has been shown to strengthen cell–cell adhesion [18].

We have recently shown the association between aberrant fucosylation and aggressive prostate cancer [19,20]. Using prostate cancer models, we have shown that overexpression of FUT8 was sufficient to transform the androgen-dependent LAPC4 prostate cancer cells into androgen-resistant cells [19]. Similarly, we demonstrated a significant correlation between FUT8 expression and Gleason grade [20]. Our studies further supported the role of FUT8 in CRPC [19]. In this study, we tried to understand how FUT8 overexpression regulates castration-resistant mechanisms in prostate cancer cells. Using a comprehensive proteomic approach, paired with the molecular characterization of FUT8 in prostate cancer cells, we were able to identify mechanisms in which prostate cancer cells alter and modify cellular proteins which help overcome steroid-dependent hormone signaling through cell surface receptors via hyper-glycosylation.

## 2. Results 

### 2.1. Characterization of FUT8 Expressing Prostate Cancer Cells using LC MS/MS Mass Spectrometry

We have previously shown that castration or androgen ablation in prostate cancer cells induced overexpression of FUT8 [19]. To further understand the role of FUT8 in the development of castration-resistant phenotypes, we developed a FUT8 overexpression LNCaP cell line model for comprehensive proteomic analysis. Briefly, protein lysate from LNCaP control, LNCaP-FUT8, LNCaP-95, and PC3 cells were prepared as shown in schematic Figure 1A. Equal amounts of tryptic digested peptides were subjected to tandem mass tag (TMT) labeling followed by fractionation, and subsequent PTMs enrichment to facilitate global, phospho-, and intact glycoproteomic (IGP) analysis as described in the Materials and Method Section. Global proteomic analysis resulted in the identification of 7303 proteins, while phosphoproteomic and IGP analyses resulted in the identification of 20,228 phosphopeptides and 39,039 intact glycosylated peptides, respectively. Using 2-fold change as the cutoff between the LNCaP-Ctr and LNCaP-FUT8, LNCaP-95, and PC3 a density distribution was plotted to evaluate the proteome changes across the cell lines. As shown in Figure 1B, the relative abundance of proteins and peptides identified in our proteomics analysis demonstrated a normal distribution pattern, with the maximum change of 16-fold in either directions compared to the median of LNCaP Ctr cells. A similar pattern between the glycosylated and phosphorylated peptides were also observed between the control vs. LNCaP-FUT8, LNCaP-95, and PC3 cell lines (Figure 1B). To further stratify the relative changes (2-fold and above) among proteins across the different cell lines, we plotted the differentially expressed proteins between LNCaP-FUT8/LNCaP Ctr, PC3/LNCaP Ctr, and LNCaP-95/LNCaP Ctr in a Venn diagram (Figure 1C). Our results demonstrated approximately 7.7% overlap of overexpressed proteins between the LNCaP-FUT8 and the androgen-ablated LNCaP-95 cells. Similarly, 5.7% were commonly overexpressed between the LNCaP-95 and PC3 cell lines (Figure 1C). A similar trend was observed for proteins downregulated with ~10%, downregulated between LNCaP-FUT8 and PC3 cells, and 6.5% commonly between LNCaP-FUT8 and LNCaP-95 cells (6.5%) analysis (Figure 1D). 

To delineate the mechanisms and cellular processes which might be differentially regulated between LNCaP Ctr and FUT8 overexpressing LNCaP-FUT8 cells and the two other prostate cancer cells (LNCaP-95 and the aggressive prostate cancer PC3 cells), we performed pathway analysis of the differentially expressed proteins using DAVID (Database for annotation, visualization, and integrated discovery). As shown in Figure 1C, proteins involved in cellular processes ranging from cytokinesis, DNA replication to endoplasmic reticulum were upregulated in LNCaP-FUT8, LNCaP-95, and PC3 cells compared to the LNCaP Ctr cells. Similarly, many downregulated proteins that were observed between the FUT8 overexpressed cells were in pathways related to peroxisome, fatty acid degradation, lysosome biogenesis, and oxidative phosphorylation (Figure 1D). 

### 2.2. Cell Surface Receptors and FUT8 Overexpression

Using the global proteomics data, a heat map was generated to show the differentially expressed proteins between the cell lines (LNCaP Ctr vs. LNCaP-FUT8, LNCaP95, or PC3 cells). As shown in Figure 2A, a significant portion (one third) of these proteins demonstrated either overexpression or suppression (Figure 2A and Appendix A). Interestingly, we identified the epidermal growth factor receptor (EGFR) as one of the candidates that was significantly changed across the LNCaP-FUT8, LNCaP-95, and PC3 cell lines compared to the LNCaP Ctr (Figure 2A). 

Several cell surface receptors including the tumor necrosis factor receptor superfamily member 10B (TNFRSF10B), the adiponectin receptor (AdipoR1), the activin A receptor type 1B (ACVR1B), the transferrin receptor protein 1 (TfR1), the epidermal growth factor receptor (EGFR), the insulin-like growth factor 1 (IGF-1), and the very low density lipoprotein (VLDL) receptor were among the few that have shown at least 4-fold overexpression in LNCaP-FUT8 cells compared to the LNCaP-Ctr (and Appendix A). To determine how FUT8 overexpression impacted EGFR signaling, we examined our phosphoproteomics results to identify differentially expressed phosphopeptides (1.35-fold change) between LNCaP-FUT8 and LNCaP-Ctr cells. Mapping these differentially expressed phosphoproteins using the Ingenuity Pathway Analysis bioinformatics tool, we found a number of proteins in the EGFR pathway including GRBs, SRC, MEKK1, MEK1, JAK1, and PKC were identified to be phosphorylated in the EGFR signaling cascade in LNCaP-FUT8 cells when compared to the wildtype control LNCaP cells (Figure 2B–D). Phosphorylated peptides that were differentially expressed between the control and LNCaP-FUT8, LNCaP-95, and PC3 were plotted as a heat map. As shown in Figure 2E and Appendix AB, five of the identified phosphorylated EGFR peptides were found to be overexpressed in LNCaP-FUT8 compared to the LNCaP-Ctr cells (Figure 2E). 

The phosphorylation of EGFR in FUT8-transformed cells was further validated by Western blot analysis in wildtype and FUT8 overexpressing LNCaP cells after 2 h of treatment with EGF or DMSO (Appendix AA). In parallel, to understand the effect of FUT8 on EGFR expression, LNCaP cells were stably selected to express four different shRNA constructs against FUT8 gene, and were analyzed for EGFR expression using Western blot analysis (Appendix AC). Data from this experiment indicated that reduced expression of FUT8 had a marginal effect on total EGFR expression (~12%) when compared to the wildtype. In contrast, overexpression of FUT8 was responsible for more than 2-fold increase of EGFR in LNCaP-FUT8 cells (Appendix AC). We have also evaluated the intact glycosylated protein related changes among the cells (Figure 2F) and have identified several intact glycosylated peptides from the EGFR protein that were differentially expressed between the control and the LNCaP-FUT8, LNCaP-95, and PC3 cells (Figure 2G and Appendix A). The different fucosylated EGFR peptides and their glycoforms identified between cells lines were plotted as log2 fold change relative to the wildtype LNCaP cells, demonstrating that overexpression of FUT8 in LNCaP-FUT8 cells contribute to the higher fucosylated levels of EGFR peptides, comparable to the levels observed in the aggressive PC3 cell line. (Figure 2H and Appendix A). 

The FUT8 protein is a member of glycosyltransferase enzymes which is responsible for the core fucosylation of secretory or cell membrane protein [21]. In order to investigate whether FUT8 expression was colocalized with the EGFR protein in prostate cancer cells, we performed an immunofluorescence (IF) colocalization study using confocal microscopy. As shown in Figure 2I, LNCaP-wt, 22RV1, LNCaP-95, and PC3 cells were double stained with FUT8 (FITC) and EGFR (TRITC) while the DNA was labeled with DAPI. Little to no colocalization of EGFR and FUT8 was observed in LNCaP wt and 22RV1 cells, while on the other hand, a strong colocalization signal as indicated by white arrows (yellow staining) were observed in PC3 cells and some in LNCaP-95 cells supporting the increased abundance of fucosylated EGFR glycopeptides observed in our intact glycoproteomics data (IGP). We also evaluated the LNCaP-FUT8 cells that were stably selected to express FUT8 using IF and found a strong costaining of FUT8 and EGFR confirming the role of fucosylation of EGFR in LNCaP-FUT8 cells (Figure 2I–K). 

### 2.3. Overexpression of FUT8 Suppress the PSA Production and Increase Drug Resistance in Prostate Cancer Cells

Activation of EGFR by the epidermal growth factor (EGF) has been shown to promote prostate cancer growth, while suppressing the PSA production and secretion in the EGF stimulated cells [22]. Several studies have indicated the involvement of growth factors and cytokines in the transformation of prostate cancer cells. However, no report has ever evaluated the involvement of the glycosylated related enzymes in the transformation processes that could lead to the signaling pathway switch in prostate cancer. To determine whether FUT8 overexpression might be responsible in overcoming mechanisms of castration-induced death by substituting the nuclear receptor AR-dependent pathway by the cell surface-dependent pathway. To understand these mechanisms, we first performed Western blot analyses on the LNCaP Ctr and LNCaP-FUT8 cells. As shown in Figure 3A, the LNCaP-FUT8 cells that were stably selected to overexpress FUT8 had higher expression levels of the EGFR protein when compared to the wildtype control. Interestingly, the prostate specific antigen (PSA) expression in the FUT8 overexpressing LNCaP cells was substantially reduced when compared to the wildtype. This suppression of PSA expression was independent of the androgen receptor (AR) which was slightly upregulated in the FUT8 overexpressing LNCaP-FUT8 cells. In order to rule out the possibility that overexpression of AR in FUT8 cells was active in terms of the PSA production, we measured the activity of the AR by using the AR firefly reporter assay as previously described [23,24] and found that FUT8 overexpression significantly suppressed the AR activity in these cells (Appendix AA). 

To better understand the inhibitory mechanisms of FUT8 on PSA secretion in prostate cancer cells, we utilized the clinical PSA ELISA assay to quantify the total and free PSA. Briefly, LNCaP wildtype or LNCaP-FUT8 cells that were stably selected to express FUT8 were plated in regular media that contains 10% FBS or charcoal stripped serum (CSS) containing media for three days. At the end of the experiment, total PSA and free PSA were measured in both cell lysates and in conditioned media. As shown in Figure 3B,C, LNCaP-FUT8 cells have lower levels of total and free PSA in both the cell lysate and cell media. These results further demonstrated that overexpression of FUT8 is inhibiting the production and release of PSA in prostate cancer cells (Figure 3B,C). To further elaborate the role of FUT8 in EGFR targeting, we next asked, whether overexpression of FUT8 in LNCaP cells might be contributing to the resistant mechanisms of gefitinib-induced cell death. Therefore, we performed the MTS assay by placing an equal number of LNCaP Ctr and LNCaP-FUT8 cells in a complete or charcoal stripped serum (CSS) containing media one day before the Gefitinib (0.5, 2.5, 5, and 10 uM) or DMSO treatment. By the end of 120 h, the MTS assay was performed to observe and compare the proliferation differences between the two cell lines. As shown in Figure 3D, LNCaP Ctr cells that were treated with Gefitin-b showed a significant growth inhibition when compared to the LNCaP-FUT8 cells. On the other hand, using the CSS (charcoal stripped serum) containing media, which suppressed the proliferation of LNCaP Ctr cells did not affect the proliferation of LNCaP-FUT8 cells even in the presence of different concentrations of Gefitinib (Figure 3D) suggesting a role of FUT8 in gefitinib and androgen-resistant mechanisms in prostate cancer cells. One possibility of LNCaP-FUT8 resistance to lower gefitinib concentration might be, that the FUT8 overexpressing cell line might switch back to the AR signaling, when the EGFR receptor was targeted. In order to evaluate our hypothesis, we used a combined blockage approach for the EGFR and AR pathway and evaluated the cell cytotoxicity using the MTS assay. Briefly, an equal number of LNCaP Ctr or LNCaP-FUT8 cells were treated with EGF or gefitinib (5 uM) or bicalutamide (10 uM) or in combination for a total of three days. After 72 h, cell viability was measured via the MTS assay. As shown in Figure 3F, compared to the wildtype control, LNCaP-FUT8 cells that were treated either with EGF (20 ng/mL), gefitinib (5 µM), or bicalutamide (10 µM) alone or in combination showed significant resistance to these treatments suggesting the role of other proteins and other signaling that were contributing to the resistance mechanisms. Further studies will be needed to understand how FUT8 overexpression might be driving multiple drugs-resistant mechanisms to support prostate cancer cell growth (Figure 3E).

### 2.4. Overexpression of FUT8 Promotes DNA Replication and Cell Cycle Progression 

Our data demonstrated the association of FUT8 and EGFR expression in LNCaP-FUT8 cells similarly, we showed that overexpression of FUT8 suppress the AR-dependent PSA production and resist gefitinib-induced cell death. Next, we explored whether FUT8 overexpression might act as a driver for DNA replication and cell cycle progression. To determine what specific DNA replication and cell cycle progression proteins were involved in LNCaP-FUT8 proliferation, we interrogated the globally acquired prostate cancer cells proteomic data. Using the 1.5-fold differentially expressed proteins between the wildtype LNCaP and LNCaP-FUT8 cells by subjecting them to the KEGG pathway analysis [19], we found a set of 21 differentially expressed DNA replication complex protein including the DNA polymerase α- primase complex and DNA polymerase epsilon complex 3 and 4 proteins that were overexpressed in FUT8 overexpressing LNCaP cells (Figure 4A, Appendix A). Similarly, we found some of the cell cycle proteins, including all members of the minichromosome maintenance complex proteins of the G1 cell phase cycle, the PCNA (proliferation cell nuclear antigen), and the CDK2 proteins (Figure 4B, Appendix A) were identified to be overexpressed in LNCaP-FUT8 cells compared to the wildtype control. The interaction between the PCNA and Cdk2 has been shown in co-immunoprecipitated complexes (PCNA-Cdk2) or together with cyclin A [25], which is required for the DNA synthesis to drive the cell through G1 into the S phase cell cycle. Our data support the notion of these co-expressing proteins which were overexpressed (~2-fold) in the LNCaP-FUT8 cells compared to the wildtype control (Figure 3C,D). 

### 2.5. Regulation of EGFR Expression by FUT8

The interactions between androgen receptor (AR) and testosterone have been extensively shown to cause upregulation of androgen specific targeted genes and downregulation of others [26,27]. Among others, EGFR is one of the genes that are downregulated by androgens and overexpressed after treating with anti-androgen therapy (bicalutamide) [28] (Appendix A). In order to understand whether androgen resistance or independence might be driving EGFR expression, we performed qRT-PCR and Western blot analysis (Figure 5A,B) using five different prostate cancer cell lines (LNCaP, C4-2, DU145, PC3, and LNCaP-FUT8) to evaluate EGFR expression. Interestingly, we found higher expression of EGFR in cells that were either androgen independent (C4-2) or do not express the AR (DU145 and PC3), as well as the LNCaP-FUT8 cells (Figure 5A,B). We used the same cell lines to evaluate the FUT8 expression and found a similar pattern of overexpression of FUT8 mRNA and protein expression in cell lines that were either androgen independent (C4-2) or resistant to testosterone (DU145 and PC3) (Figure 5C,D) suggesting a positive association between FUT8 and EGFR protein expression in androgen-resistant prostate cancer cell lines. 

### 2.6. Castration-Induced FUT8 Overexpression Correlates with EGFR Expression in Xenograft Model

In our in vitro data, we have demonstrated that overexpression of FUT8 was regulating the expression of EGFR in prostate cancer cells (Figure 2A), which was responsible for the androgen-resistant mechanism (Figure 3B,E). In order to confirm that FUT8-induction was in fact driving the overexpression of EGFR in vivo, we inoculated the LNCaP cells as xenografts on the dorsal lateral flanks of athymic nude mice. When the tumor volume reached around 1 cm^3^, animals were divided into two groups. Castration was performed in one group of animals by bilateral orchiectomy, while the control group were shame-operated under anesthesia to localize sex organs but without orchiectomy. At the end of the experiment (five weeks after castration), all animals were sacrificed, and tumors were removed for immunohistochemical studies. Tissues were stained for FUT8, EGFR, and LCA lectin. As shown in Figure 6, tumor sections obtained from the castrated animals that were stained with FUT8 antibody, demonstrated a significant higher expression of FUT8 compared to the tissue sections from uncastrated controls, suggesting the induction of FUT8 under androgen-restricted conditions. A similar trend of EGFR overexpression was also observed in castrated conditions confirming our in vitro data. Lectins have been shown to display selection specificity for certain glycoproteins. To assess whether overexpression of FUT8 enzyme in these cells were also impacting the overall fucosylation status of the tumor tissues, we stained these tumor sections for both the castrated and noncastrated xenograft tumors using LCA lectin. LCA lectin has been previously shown to bind with high selectivity and specificity to the core fucosylated glycopeptides [29]. As shown in Figure 6, a higher staining intensity for LCA lectin was observed in tumors from castrated mice, confirming that FUT8 overexpression was responsible for higher core fucosylated protein in castrated xenograft tumors. Taken together, these data confirm our in vitro qPCR and LC–MS/MS protein expression data that FUT8 overexpression might be driving the castration-dependent EGFR regulation in prostate cancer. 

## 3. Discussion

Mechanisms of androgen resistance is one of the underlying causes for recurrence in advanced prostate cancer. In our study, we tried to understand how prostate tumor cells adjust to the androgen deprivation and how the cancer cells manipulate its proteome in order to compensate therapeutic assaults. In addition to mutations and epigenetic regulation of genes, the post-translational regulation at the proteome levels is another mechanism that might play some role in tumor escape from the stressed or testosterone deserted environment. We have previously demonstrated the androgen ablation in prostate cancer-induced FUT8 expression. Similarly, our initial studies with FUT8 in prostate cancer specimens demonstrated a strong correlation between FUT8 expression and aggressive prostate cancer. However, it was unclear how prostate cancer cells manage to escape cell death by upregulating the FUT8 gene expression in the androgen-depleted environment and if testosterone (R1881) was negatively affecting the FUT8 levels. Our qRT-PCR data confirmed the inhibitory activity of R1881 on FUT8 expression. In this study, we took a comprehensive proteomic approach by studying the global, phospho-, and glyco-proteomics profiles of FUT8 overexpressing prostate cancer cells. The FUT8 enzyme belongs to the family of N-linked glycosyltransferases and is responsible for the transfer of fucose from the GDP-fucose to the core-GlcNAc of the N-linked glycoproteins. While all of the N-linked glycosylation are known to occur on protein that are either in the secretory pathways or the membrane bound receptor proteins, we became interested to explore whether there was a differential regulation for cell surface receptor in cell lines that were overexpressing the FUT8 enzyme. Analyzing the global LC–MS/MS proteomic data, we compared the expression profiles of the cell surface receptor between the wildtype and LNCaP-FUT8 cell lines. Interestingly, we found a significant higher level of EGFR protein in LNCaP-FUT8 cells, with a similar overexpression of EGFR protein also observed in the PC3 cells. Using a cutoff of 1.5-fold changed between LNCaP Ctr and LNCaP-FUT8 that was selected to overexpressed FUT8, we subjected the identified phosphorylated proteins to the ingenuity pathway analysis (IPA) and found that many of the EGFR downstream signaling proteins were upregulated in the LNCaP-FUT8 cells. These data indicated that overexpression of FUT8 was driving the cell growth and proliferation by engaging the RAS, RAF, MAPK, and JNK1 proteins in the LNCaP-FUT8 prostate cancer cell model, representative of prostate cancer transformation via cell signaling tropism from AR to EGFR signaling cascades. 

The first line of intervention for disseminated advanced prostate cancer entails the use of AR-directed therapies. These options, however, always fail in the face of resistant mechanisms. Several mechanisms have been proposed and investigated to understand these resistant phenotypes. In this study, we evaluated the role of FUT8, a glycosylated related enzyme as a master regulator to be involved in the transformation of prostate cancer cells from nuclear receptor AR-dependent signaling to the cell surface including the EGFR signaling. Targeting the LNCaP-FUT8 and control cells with different concentrations of the EGFR inhibitor showed resistance to the cell death in LNCaP-FUT8 cells when compared to the wildtype control, and may suggest that FUT8 overexpression might be responsible for activation of several other signaling cascades that can overcome the EGFR inhibition. Similarly, other members of the Erb family that are capable of forming heterodimers with EGFR or posttranslational modifications and overexpression of other cell surface receptors, including IGF-1, identified in our proteomics data (Appendix A) have been shown to play anti-apoptotic roles in prostate cancer [30]. In addition, overexpression of IGF-1 has been linked to the resistance mechanisms of Gefitinib-induced cell death [31]. 

Utilizing the proteomics approaches, we demonstrated how FUT8 overexpression can regulate the EGFR expression through glycosylation and phosphorylation and the downstream signaling cascade. We further confirmed these studies in castrated xenograft models by showing a significant association of EGFR and FUT8 expression. Our results also confirm previous studies showing a negative association between the PSA and EGFR signaling [22]. Based on our knowledge, this is the first report that describes how FUT8 in an androgen-depleted condition might be driving the expression of EGFR and rescuing prostate cancer cells from depleted androgen-induced cell death. Our findings have clinical implications and support to explore the combination regimen of FUT8 inhibitors and anti-androgen in FUT8 overexpressing aggressive prostate disease, but more importantly, our studies propose evaluating FUT8 overexpression as a surrogate biomarker for biochemical recurrence. Studies are currently underway in our laboratory to retrospectively evaluate FUT8 expression in tumor tissue microarrays (TMAs) cohorts to interrogate our hypothesis.

## 4. Materials and Methods

### 4.1. Cell Lines and Reagents

Prostate cancer LNCaP (an androgen-responsive cell line with mutant AR), 22RV1 (a CRPC cell line), and PC3 (androgen-independent) cell lines that were previously obtained from the American Type Culture Collection (Manassas, VA, USA) were maintained in RPMI-1640 supplemented with 10% FBS according to the ATCC recommendations. LNCaP-95 cells (androgen-independent cell line) were a gift from Dr. Alan Meeker (Johns Hopkins School of Medicine) that were previously described [32]. The LNCaP-FUT8 and LNCaP Ctr cell lines were generated from the wildtype LNCaP cells by transfecting plasmid DNA carrying FUT8 gene or empty vector (Origene, Maryland). Stable cell lines were generated and maintained in 100 ug/mL Geneticin (G418). Primary rabbit monoclonal antibodies for AR (AR 441, dilution 1:1000) and polyclonal AR (N-20, dilution 1:1000) were from Santa Cruz Biotech, (Santa Cruz, CA, USA), Polyclonal EGF Receptor (D38B1) XP antibody was obtained from Cell Signaling Technology, Inc (Danvers, MA, USA). The polyclonal FUT8 antibody was from R&D systems (Minneapolis, MN, USA) and a kind gift from Drs Ruey-Bing Yang and Cheng-Fen Tu (Institute of Biomedical Sciences, Academia Sinica, Taipei). Similarly, mouse monoclonal β Actin (1:25,000), the anti-mouse and the anti-rabbit IgG HRP-conjugated (1:20,000) were from Sigma-Aldrich (St. Louis, MO, USA). The anti-sheep IgG HRP-conjugated (1:20,000) was from ThermoFisher Scientific (Grand Island, NY, USA). The majority of all other chemical reagents and compounds were ordered from Sigma, unless otherwise specified.

### 4.2. Western Blot Analysis

Western blotting was done as previously described [33,34] to confirm some of the protein identified in our Mass-Spec analysis. Briefly, cells were washed with 1 × PBS and resuspended with five volumes of a cold lysis buffer (50 mM Tris-HCl, pH 7.5, 250 mM NaCl, 5 mM EDTA, 50 mM NaF, 0.5% NP-40) supplemented with protease inhibitor cocktail (Roche, Indianapolis, IN). The cell lysate was incubated on ice for 30 min and then centrifuged for 10 min at 4 °C. Equal amounts of proteins were separated by SDS-PAGE, and the resolved proteins were then transferred onto a nitrocellulose membrane. After blocking with 5% nonfat milk in TBST overnight at 4 °C, the blot was incubated with primary antibody at 1 h at room temperature. Then, the membrane was probed with a HRP-conjugated secondary antibody for 1 h and developed (ECL-Plus system, Amersham Pharmacia, Piscataway, NJ, USA) using the manufacturer’s protocol.

### 4.3. Protein and Peptide Extraction from Cells for Proteomic Analysis

The cell pellets for LNCaP Ctr or LNCaP-FUT8, LNCaP-95, and PC3 cells from T-75cm flasks was first denatured in 1 mL of 8 M urea and 0.4 M NH4HCO3 and sonicated thoroughly. The protein concentrations were measured using a BCA protein assay kit (Thermo Fisher Scientific Inc). Isolated proteins were then reduced by incubating in 120 mM Tris (2-carboxyethyl) phosphine for 30 min and alkylated by addition of 160 mM iodoacetamide at room temperature for 30 min in the dark. The sample was diluted with a buffer (100 mM Tris-HCl, pH 7.5) containing 0.5 ug/uL trypsin and incubated at 37 °C overnight. The digested proteins were checked for completion of trypsin digestion using SDS-PAGE. Peptides were purified with C-18 desalting columns and dried using the SpeedVac. 

### 4.4. TMT Labeling of Tryptic Peptides from Prostate Cancer Cell Lines

Four hundred micrograms (400 ug) of digested tryptic peptides from each cell lysate was labeled with an Amine-Reactive tandem mass tag reagent (TMT) according to the manufacturer’s protocol (Thermo Scientific). Briefly, protein was solubilized in TEAB (90 mM) together with 0.5% SDS. Digested peptides were reduced with tris (2-carboxyethyl) phosphine for 1 h at 55 °C followed by alkylation with 17 mM iodoacetamide for 30 min in the dark with gentle shaking. The TMT labels were reconstituted prior to labeling in 24 uL of DMSO and added to each sample for labeling at room temperature for at least 1 h. Each reaction was quenched by the addition of 5% hydroxylamine. After 15–20 min the samples were combined together in a single 1.5 ml tube. The labeled mixed TMT peptides were purified by a SCX column. The labeled peptides were dried and resuspended into 0.4% acetic acid solution prior to fractionation for mass spectrometry analysis.

### 4.5. Chromatography Fractionation

The TMT labeled tryptic peptides were fractionated into 24 fractions by basic reverse phase liquid chromatography (bRPLC) using a 1200 Infinity LC system (Agilent Technology, Santa Clara, CA, USA) equipped with a 4.6 × 100 mm BEH120 C-18 column (Waters, Milford, MA, USA). The mobile-phase A consisted of 10 mM ammonium formate (pH 10) and B consisted of 10 mM ammonium formate and 90% acetonitrile (pH 10). A total of 50 µg peptides were fractionated using the following linear gradient: From 0% to 2% B in 10 min, from 2% to 8% B in 5 min, from 8% to 35% B in 85 min, from 35% to 95% B in 5 min, and then held at 95% B for an additional 15 min. Peptides were detected at 215 nm and 96 fractions were collected along with the LC separation in a time-based mode from 16 to 112 min. The separated peptides in 96-wells were concatenated into 24 fractions by combining four wells into one sample, such as 1, 25, 49, and 73 as fraction one; 2, 26, 50, and 74 as fraction two; etc. Then, the peptides were dried in a speed-vacuum and stored at −80 °C until liquid chromatography-tandem mass spectrometry (LC–MS/MS) analysis

### 4.6. Mass Spectrometric Analysis 

Five percent (5%) of each fraction was utilized for global protein mass spectrometry analysis. The remaining 95% of the fractioned peptides were run via the IMAC method as recently described by us [35] for phosphorylated enrichment while the flow through was utilized for intact glycopeptide enrichment using MAX extraction cartages [29]. Samples were analyzed in a QE quadrupole Orbitrap mass spectrometer (Thermo Scientific) coupled to an Ultimate 3000 chromatography system (Dionex). Reversed-phase chromatographic separation was carried out on a 75 um × 50 cm C18 PepMap RSLC column (thermos Scientific) protected by 5 mm guard column C-18 Nanocolumn, 3 uM bead size, 100 Å pore size (Dionex; #164261) with a linear gradient of 5%–50% solvent B (99.9% ACN/0.1% FA). The mass spectrometer was operated in the data-dependent mode to automatically switch between Orbitrap MS and iontrap MS/MS acquisition. Survey full scan MS spectra (from m/z 400 to 2000) were acquired in the Orbitrap with a resolution of 60,000 at m/z 400 and an AGC (automatic gain control) target value of 1x 10^6^ ions. For identification of TMT labeled peptides, the three most abundant ions were selected for CID, and then the same precursor ions triggered higher energy collisional dissociation (HCD) fragmentation. HCD normalized collision energy was set to 50% and fragmentation ions were detected in the Orbitrap at a resolution of 7500. CID fragmentation ions were detected in the LTQ ion trap. Target ions that had been selected for MS/MS were dynamically excluded for 60 s. For accurate mass measurement, the lock mass option was enabled using the poly-dimethylcyclosiloxane ion (m/z 455.120025) as an internal calibrant. For peptide identification, raw data files produced in the Xcalibur software (Thermo Scientific) were processed in Proteome Discoverer V1.2 (Thermo Scientific) prior to Macsot searching against the UniProtKB/Swiss-Prot database (Release 2010_08 of 17-Jul-10: 518,415 entries). For searching criteria, taxonomy was selected to human; the MS tolerance was set to ± 20 ppm, and the MS/MS tolerance to 0.8 Da. Two missed cleavages were allowed and carbamidomethylation of cysteines was set as fixed modification. TMT modification of peptide N-termini and lysine residues, methionine oxidation, deamidated (NQ), and Gln-pyroGlu (N-termini Q) were set as variable modifications. Search result filters were selected as follows; only peptides with a score >20 and below the Mascot significance threshold filter of p = 0.05 were included and single peptide identifications required a score equal to or above the Mascot identity threshold. Protein grouping was enabled such that when a set of peptides in one protein were equal to or completely contained within the set of peptides of another protein, the two proteins were put together in a protein group. Quantitative information calculated from reporter ion intensities were only accepted.

### 4.7. Protein Expression Data Analysis

MS/MS Data and Differential Expression Analysis: We searched our tandem mass spectrometry derived raw data against the RefSeq protein database using the SEQUEST(R) search engine in Proteome Discoverer v1.4. We specified oxidation of methionine, carbarmidomethylation of cysteine, N-terminal TMT modification as fixed residue modifications. We specified lysine (K) and tyrosine (Y) TMT modifications as dynamic modifications. Peptide identification false discovery rate (FDR) was specified as 0.01. Parsimonious protein grouping was specified to allow at least two peptides per protein. High confidence PSMs (i.e., peptide spectrum match better than prespecified false discovery rate cutoff) were used for protein grouping. Peptide (peptide spectrum match) and protein quantifications were based on ratios of TMT reporter ions; 126, 127C, 127N, and 128N. Our specified reporter ion quantification LNCaP-FUT8/LNCaP Ctr was 127C/126. The PC3/LNCaP Ctr ratio was (127N/126) and the LNCaP-95/LNCaP Ctr was 128C/126. The proteome discoverer (PD) also reports coefficient of variations (CVs) for unique peptide spectrum matches which was set at the acceptable limit at less than or equal to 30%. We filtered out peptide spectrum matches and associated protein with reporter ion ratio CVs greater than 30%.

### 4.8. qRT-PCR Analysis

mRNA (1 μg) from wildtype LNCaP or LNCaP-FUT8 and the androgen-resistant LNCaP-95 cells were each reverse transcribed using the QuantiTect Reverse Transcription kit (Qiagen). Syber green–based real-time qRT-PCR was performed using SYBR GreenER qPCR SuperMix (Invitrogen) according to the manufacturer’s instructions. Q-PCR primer pairs (2 nmol mix) for FUT8 CAT#: HP231910, beta Actin (CAT#: HP204660) and EGFR (CAT#: HP208404) were bought from Origene Inc (Maryland, USA). Standard curves were generated by serial dilution of each sample, and the relative amount of FUT8 or EGFR mRNA in the cell lines was normalized to ATCB mRNA using the following specific set of primers:FUT8-Forward: GACAGAACTGGTTCAGCGGAGAFUT8-Reverse: GCAGTAGACCACATGATGGAGCActin-Forward: CACCATTGGCAATGAGCGGTTCActin-Reverse: AGGTCTTTGCGGATGTCCACGTEGFR-Forward: AACACCCTGGTCTGGAAGTACGEGFR-Reverse: TCGTTGGACAGCCTTCAAGACC

### 4.9. Xenograft Animal Models 

LNCaP wildtype cells at a density of 1x 10^6^ in PBS were mixed in 1:1 ratio with 1x Matrigel (BD biosciences) and implanted into the dorsal flanks of Athymic nude male mice. Once tumors were established and reached tumor volume of ~1 cm^3^, animals were divided into two groups by total tumor volume with five animals per group. Since no data were available to evaluate the effect of castration on the LAPC4 xenografts, as well as the dichotomous outcome of the FUT8 staining between the groups, we chose to have five animals per group to get enough power (76.6%) that achieve 1.5 standard deviation differences, assuming equal variability among each group. Castration was achieved by orchiectomy. Mice in the control group were shame-operated under anesthesia. Animal studies were carried out in compliance with the USA Public Health Service Policy as approved by the Institutional Animal Care and Use Committee. Animals were monitored every other day for any signs of cachexia and distress. No animals were excluded from the analysis. Investigators involved in the animal experiment were not blinded to the treatment. At the end of the experiment (five weeks after castration) all animals were sacrificed, and tumors were removed. Two tumors from each group were used for immunohistochemistry using the FUT8, AR, Biotinylated LCA Lectin (Vector Labs) and EGFR antibodies (Minneapolis, MN, USA).

### 4.10. Immuno-Histochemical (IHC) Staining

IHC staining was performed using antibodies FUT8, AR, and EGFR at the dilution of 1:500 biotinylated LCA lectin was used at 1:1000 dilution. Paraffin embedded tissues slides were deparaffinized and rehydrated. Antigen retrieval was performed using the Target Antigen Retrieval Solution (DAKO, CA) by boiling inside the steamer pot for 25–30 min. The endogenous peroxidase was blocked using a 3% hydrogen peroxide solution followed by serum blocking solution from the DAKO. Primary antibodies diluted in the antibody dilution buffer (DAKO, CA, USA) were incubated for 1 h before incubating the slides with primary antibody overnight at 4 °C. The secondary antibody ready to use the polymer with anti-mouse/rabbit (DAKO, CA, USA), was incubated at room temperature for 30 min, and bound peroxidase was detected using the DAB (DAKO, CA, USA). All IHC slides were counterstained with hematoxylin. For tissue morphological analysis, microscopic images were examined under 10x or 40x magnification using an inverted light microscope (Zeiss, NY, USA).

### 4.11. Fluorescence Microscopy

The immunofluorescence method was performed using the LNCaP, 22RV1, LNCaP-95, PC3, and LNCaP-FUT8 cells. Briefly, cancer cells were grown on poly-d-lysine coated glass coverslips that were fixed with 2% paraformaldehyde-PBS for 10 min. After fixation, cells were washed three times with PBS, followed by permeabilization in 0.1% Triton X-100 for 5 min for permeabilization. The fixed cells were blocked in 1% bovine serum albumin in PBS for 15 min, followed by a 1-h incubation with the primary antibodies against FUT8 or EGFR. After washing three times with TPBS, cells were then probed with secondary antibodies (rhodamine-conjugated anti-rabbit IgG or FITC-conjugated anti-mouse IgG), followed by DNA staining with DAPI for at least 10 min. Images were taken with a LSM 710 confocal microscopy (Carl Zeiss, Jena, Germany). Figures were constructed using Adobe Photoshop (Adobe Systems, CA). Colocalization of the red and green signals were quantified using the image-pro plus (Rockville, MD) to obtain the Pearson’s correlation (Rr) and overlap coefficients (R) between green and red channels according to the software guidelines. 

### 4.12. Statistical Analysis

All experiments were done in triplicate or quadruplicate and plotted with SEM or otherwise indicated. Statistical analysis was performed using Excel running on an IBM-PC compatible computer on Windows 7 operating system. Statistical comparisons for the in vitro data were analyzed by two-sided Student’s t-test. Statistical significance was defined as *p* < 0.05.

## 5. Conclusions

Based on our knowledge, this is the first report that describes how FUT8 in androgen-depleted condition might be driving the expression of EGFR and rescuing prostate cancer cells from depleted androgen-induced cell death. Our findings have clinical implications and support to explore the combination regimen of FUT8 inhibitors and anti-androgen in FUT8 overexpressing aggressive prostate disease, but more importantly, our studies propose evaluating FUT8 overexpression as a surrogate biomarker for biochemical recurrence. Studies are currently underway in our laboratory to retrospectively evaluate FUT8 expression in tumor tissue microarrays (TMAs) cohorts to interrogate our hypothesis.

## Figures and Tables

**Figure 1 cancers-12-00468-f001:**
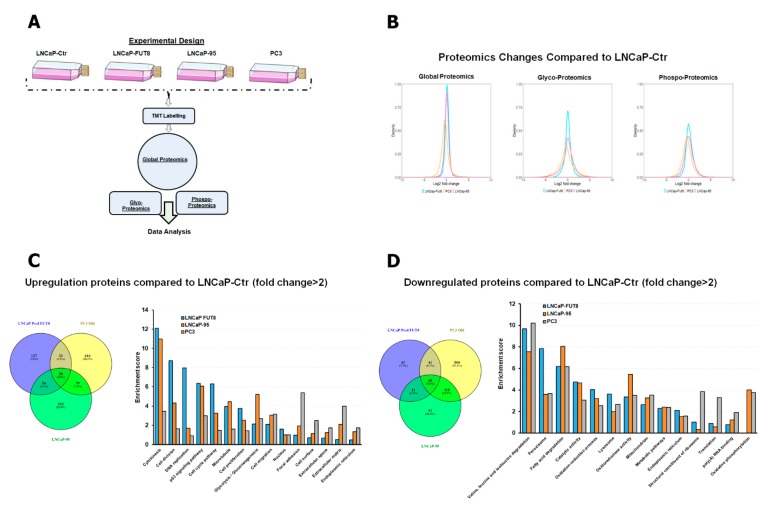
Characterization of LNCaP-FUT8 cells using LC–MS/MS: Schematics for proteomics strategies including Tandem mass tag (TMT) labeling, global proteomics, phosphoproteomics, and glycoproteomic to characterize LNCaP-FUT8 cells (**A**). Density distribution plots for the log 2-fold changed proteins ratio between LNCaP-FUT8, LNCaP-95, and PC3 to LNCaP ctr cells for global, phospho, and glycoproteins (**B**). Venn diagram showing the differentially regulated (2-fold) global protein ratios between cells lines compared to the LNCaP-Ctr. The identified proteins were subjected to the David software (Database for annotation, visualization, and integrated discovery) to facilitate the distribution of these proteins between different cellular signaling pathways that are overexpressed (**C**) or downregulated in LNCaP-FUT8, LNCaP-95, and PC3 cells (**D**).

**Figure 2 cancers-12-00468-f002:**
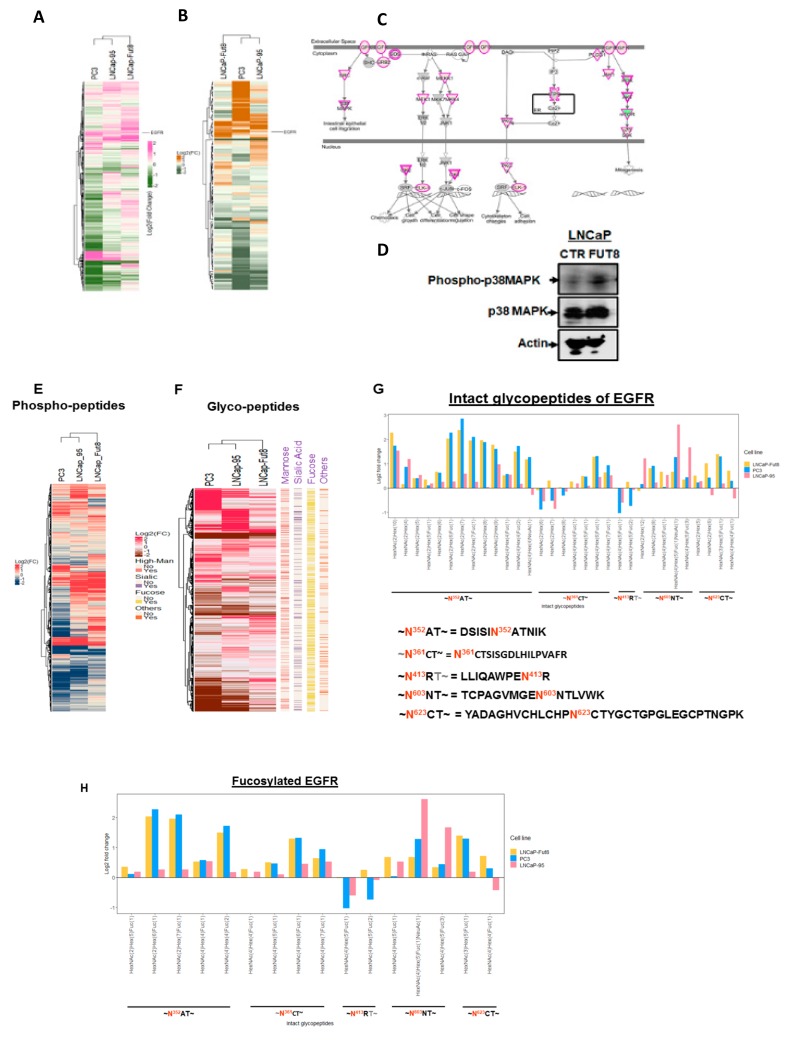
Effects of FUT8 expression on global and cell surface receptors. Heat map from the globally identified proteins (log2 fold change) (**A**) and the cell surface receptors (**B**) that were differentially expressed between prostate cancer cells (LNCaP-FUT8, LNCaP-95, and PC3) compared to the wildtype LNCaP-Ctr cells (**A**,**B**). Using a cutoff of 1.5-fold change between phosphorylated proteins in LNCaP-FUT8/LNCaP Ctr, were subjected to ingenuity pathway analysis which demonstrate the activation of epidermal growth factor receptor (EGFR) signaling pathway (**C**). Western Blot analysis for the total and phospo-p38 MAPK kinases protein (**D**). Ratios of the phosphorylated peptides for all the cell surface receptors identified in the phosphoproteomics data were plotted as a heat map to show the differential expression of phosphorylated cell surface receptors (**E**) and intact glycosylated peptides (**F**). Fold changes between the intact glycopeptides for EGFR (**G**) and the differentially expressed fucosylated glycoforms for the EGFR (**H**) identified across the cell lines. Confocal microscopy to evaluate the localization for the EGFR and FUT8 in LNCaP-WT, 22RV1, LNCaP-95, PC3, and LNCaP-FUT8 cells. White arrows indicate colocalization of the FUT8 and EGFR proteins. (**I**,**J**). Quantitation and Pearson’s correlation for the colocalization of the red and green signals between the cell lines (**K**).

**Figure 3 cancers-12-00468-f003:**
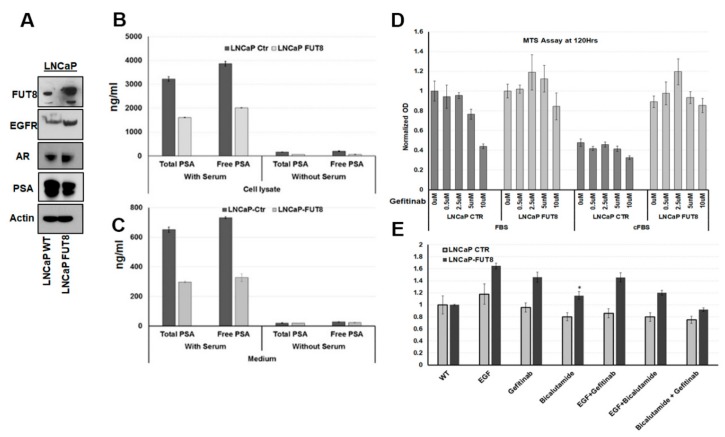
FUT8 suppresses the Prostate specific antigen (PSA) production by activating EGFR. Western blot analysis showing overexpression of EGFR proteins in LNCaP-FUT8 cells that were stably selected to express FUT8. A lower expression of PSA protein was observed in the same LNCaP-FUT8 cells when compared to the wildtype LNCaP Ctr (**A**). The Hybritech PSA ELISA assay to quantify the total and free PSA in LNCaP Ctr and LNCaP-FUT8 cell lysate (**B**) and in cultured condition media (**C**). The 3-(4,5-dimethylthiazol-2-yl)-5(3-carboxymethonyphenol)-2-(4-sulfophenyl)-2H-tetra zolium ( MTS) assay in LNCaP-FUT8 and LNCaP ctr cells that were treated with different concentrations of Gefitinib (0-10 uM) in normal and charcoal stripped cFBS containing media (**D**). MTS assay in LNCaP-FUT8 and LNCaP ctr cells that were treated with Gefitinib (5 uM or bicalutamide (10 uM) alone or in combination with both drugs for 72 h (**E**). The asterisks represent statistical difference between the two treatments at *p* < 0.05 Student’s t-test. Error bars represent ± SEM.

**Figure 4 cancers-12-00468-f004:**
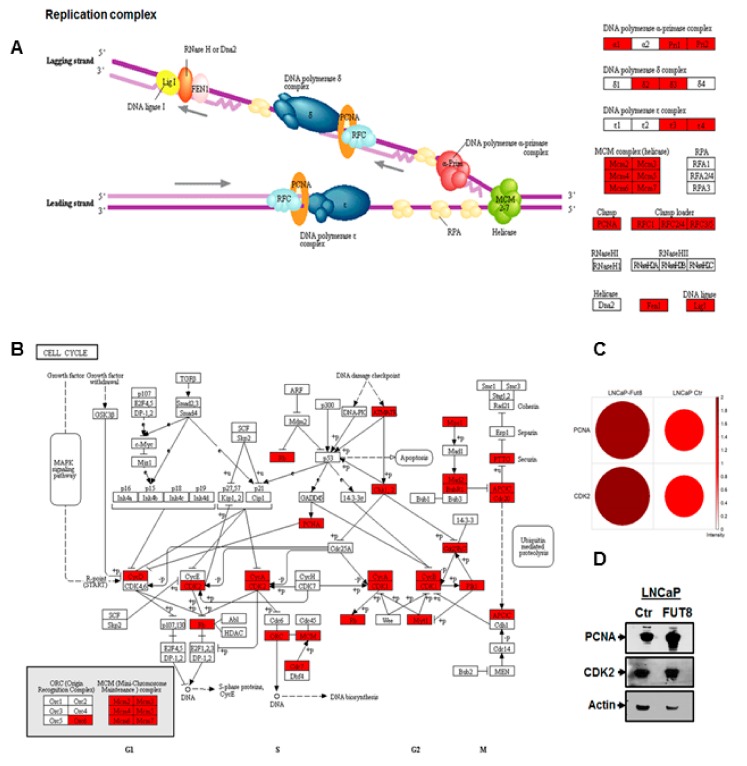
FUT8 overexpression promotes DNA replication. Differentially expressed proteins (1.5-fold upregulated proteins) between LNCaP-FUT8/LNCaP-Ctr that were identified in our global proteomics analysis were subjected to the Kyoto Encyclopedia of Genes and Genomes (KEGG) pathway analysis. Several proteins including the subunits of DNA polymerase alpha primase complex, DNA polymerase delta and epsilon along with minichromosome maintenance (MCM) and proliferation cell nuclear antigen (PCNA) were identified in LNCaP-FUT8 cells compared to the wildtype control (**A**). The same 1.5-fold upregulated proteomics data were analyzed for cell cycle analysis showing overexpression of many S phase and G2/M cell phase cycle proteins (**B**). Bubble blot showing the expression of PCNA and CDK2 expression correlation between the two LNCaP-FUT8 and LNCaP Ctr cell lines (**C**). Western Blot analysis showing PCNA and CDK2 expression between the LNCaP Ctr and LNCaP-FUT8 cell lysate. Actin was included that accounted for the amount of loading across lanes (**D**).

**Figure 5 cancers-12-00468-f005:**
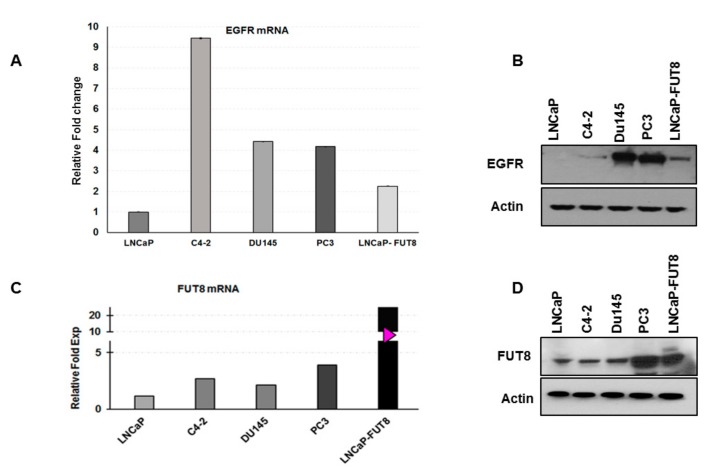
FUT8 regulates EGFR expression. The qRT-PCR and Western blot analysis of different prostate cancer cell lines showing the expression of EGFR (**A**,**B**) and FUT8 (**C**,**D**). Error bars represent ± SE of the mean. Significance was defined as the *p*-value ≤ 0.05 (**A**,**C)**.

**Figure 6 cancers-12-00468-f006:**
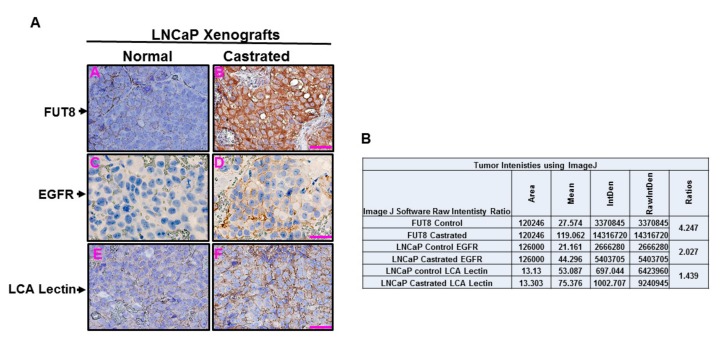
Castration-induced FUT8 expression is associated with EGFR. LNCaP cells that were inoculated as xenografts in nude mice were either castrated or mock operated after the tumor volume reached around 1 cm^3^. Five weeks after the castration all the mice were sacrificed and tumors were removed for Immunohistochemistry (IHC) staining using FUT8, EGFR, and with Lens culinaris (LCA) lectin staining. For each antibody two different tumors were stained. Images were taken using 40x magnification and quantified using ImageJ software (**A**,**B**).

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
