# Peer review of "A Comprehensive Analysis of FUT8 Overexpressing Prostate Cancer Cells Reveals the Role of EGFR in Castration Resistance"

_cancers, 2020, doi:10.3390/cancers12020468_

Round 1
Reviewer 1 Report
The authors have satisfactorily addressed the concerns that I had raised. They have provided additional data/explanation in response to my comments and have appropriately revised the manuscript.
Reviewer 2 Report
1) EGFR primer sequence missing (Fig. 5)
2) Amplicon size for Actin is 323bp. For qPCR, primers amplifying regions between 75-150bp are preferable as longer PCR product can saturate the SYBR green signal and mask the differences in the expression levels, if any. Also, the primers are not specific and bind to POTE ankyrin domain family member E (POTEE) family of mRNA with only one and two mismatches in the forward and the reverse primer, respectively.
3) Absence of phosphatase inhibitor in cell lysis buffer for WB analysis will affect the outcome on probing for phospho-proteins e.g. phospho-p38 MAPK
4) Fig.2, Page 6: Labelling of the WB image needs to be changed from "Phop38MAPK" to "Phospho-p38MAPK".
5) It is highly surprising that one is able to pick up the EGFP signal (in Fig 2I) on the cell surface of LnCaP cells by immunostaining but the lysates cant be probe for EGFR using WB (Fig. 5B). What can be the possible reason?
6) Authors are requested to show Cell cycle analysis using flow cytometry in addition to determining protein expression (Fig 4D) of proteins involved in DNA synthesis.
7) The immunostained sections used in the Normal and castrated groups do not appear to be serial sections.
8) The nuclear staining between the FUT8 and EGFR or LCA lectin and EGFR are drastically different. Is there a reason for these differences?
Author Response
EGFR primer sequence missing (Fig. 5)
The forward and reverse sequences of EGFR primers have been added in the material and methods section.
2) Amplicon size for Actin is 323bp. For qPCR, primers amplifying regions between 75-150bp are preferable as longer PCR product can saturate the SYBR green signal and mask the differences in the expression levels, if any. Also, the primers are not specific and bind to POTE ankyrin domain family member E (POTEE) family of mRNA with only one and two mismatches in the forward and the reverse primer, respectively.
We totally agree with our reviewer. We apologize for the errors. All the qPCR primers used in our studies were purchased from the OriGene Technologies Inc (Maryland). We have added the description of the primers to the material and method section and have also corrected the actin qRT PCR primer sequences which correspond to the commercially available sequences, where the length of the amplicon is ~ 135bp.
3) Absence of phosphatase inhibitor in cell lysis buffer for WB analysis will affect the outcome on probing for phospho-proteins e.g. phospho-p38 MAPK
We thanks our reviewer for pointing it out. All our proteomics protocols consist of the phosphatase inhibitors (Phosphatase Inhibitor Cocktail 2 and Phosphatase Inhibitor Cocktail 3, David et al, 2019, Zhang H et al., 2016). Similarly, Western blot analysis where phosphorylated-protein expression were performed, we have used phosphatases inhibitors together with protease inhibitor cocktail. We have now added these description to the method and material section of our re-revised manuscript.
Reference:
Clark DJ, Dhanasekaran SM, Petralia F et al., Integrated Proteogenomic Characterization of Clear Cell Renal Cell Carcinoma. Cell. 2020 Jan 9;180(1):207. doi: 10.1016/j.cell.2019.12.026 Zhang H, Liu T, Zhang Z, Payne SH, Zhang B et al., Integrated Proteogenomic Characterization of Human High-Grade Serous Ovarian Cancer. Cell. 2016 Jul 28;166(3):755-765
4) Fig.2, Page 6: Labelling of the WB image needs to be changed from "Phop38MAPK" to "Phospho-p38MAPK".
We have corrected the typo in Figure 2
5) It is highly surprising that one is able to pick up the EGFP signal (in Fig 2I) on the cell surface of LnCaP cells by immunostaining but the lysates can’t be probe for EGFR using WB (Fig. 5B). What can be the possible reason?
We appreciate the reviewer concerns, the radiograph represents western blot analysis of EGFR in several different cell lines that were exposed for shorter duration in single time (5 sec) point to avoid saturation signals in cell lines that were overexpressing EGFR protein. On the other hand Immunofluorescence microscopy was performed with different exposure times to collect maximum signals for proteins co-localization.
6) Authors are requested to show Cell cycle analysis using flow cytometry in addition to determining protein expression (Fig 4D) of proteins involved in DNA synthesis.
We very much appreciate the reviewer comments, while we understand the importance of cell cycle analysis, we believe these data will be outside the scope of this paper. In this study, we aimed to identify the altered protein expression by FUT8 using LC MS/MS proteomics and we further confirmed the changes of protein expression by western blot analysis.
7) The immunostained sections used in the Normal and castrated groups do not appear to be serial sections.
Yes, they were not serial sections. We do not have serial session since several cross sections from the paraffin embedded tissues were used for different antibodies and their optimization protocols including the EGFR in this manuscript.
8) The nuclear staining between the FUT8 and EGFR or LCA lectin and EGFR are drastically different. Is there a reason for these differences?
We thank our reviewer for the concern. The IHC staining between the EGFR and FUT8/LCA lectin were performed several weeks apart and therefore, they looked different.
Round 2
Reviewer 2 Report
The authors have addressed the concerns relating to the manuscript adequately and thus the manuscript can be considered for publication.